# WHAT CAN NEURAL NETWORKS REASON ABOUT?

**Keyulu Xu**[†], **Jingling Li**[‡], **Mozhi Zhang**[‡], **Simon S. Du**[§], **Ken-ichi Kawarabayashi**[¶],
**Stefanie Jegelka**[†]
[†]Massachusetts Institute of Technology (MIT)
[‡]University of Maryland
[§]Institute for Advanced Study (IAS)
[¶]National Institute of Informatics (NII)
{keyulu, stefje}@mit.edu

## ABSTRACT

Neural networks have succeeded in many reasoning tasks. Empirically, these tasks require specialized network structures, e.g., Graph Neural Networks (GNNs) perform well on many such tasks, but less structured networks fail. Theoretically, there is limited understanding of why and when a network structure generalizes better than others, although they have equal expressive power. In this paper, we develop a framework to characterize which reasoning tasks a network can learn well, by studying how well its computation structure aligns with the algorithmic structure of the relevant reasoning process. We formally define this algorithmic alignment and derive a sample complexity bound that decreases with better alignment. This framework offers an explanation for the empirical success of popular reasoning models, and suggests their limitations. As an example, we unify seemingly different reasoning tasks, such as intuitive physics, visual question answering, and shortest paths, via the lens of a powerful algorithmic paradigm, dynamic programming (DP). We show that GNNs align with DP and thus are expected to solve these tasks. On several reasoning tasks, our theory is supported by empirical results.

## 1 INTRODUCTION

Recently, there have been many advances in building neural networks that can learn to reason. Reasoning spans a variety of tasks, for instance, visual and text-based question answering (Johnson et al., 2017a; Weston et al., 2015; Hu et al., 2017; Fleuret et al., 2011; Antol et al., 2015), intuitive physics, i.e., predicting the time evolution of physical objects (Battaglia et al., 2016; Watters et al., 2017; Fragkiadaki et al., 2016; Chang et al., 2017), mathematical reasoning (Saxton et al., 2019; Chang et al., 2019) and visual IQ tests (Santoro et al., 2018; Zhang et al., 2019).

Curiously, neural networks that perform well in reasoning tasks usually possess specific *structures* (Santoro et al., 2017). Many successful models follow the Graph Neural Network (GNN) framework (Battaglia et al., 2018; 2016; Palm et al., 2018; Mrowca et al., 2018; Sanchez-Gonzalez et al., 2018; Janner et al., 2019). These networks explicitly model pairwise relations and recursively update each object's representation by aggregating its relations with other objects. Other computational structures, e.g., neural symbolic programs (Yi et al., 2018; Mao et al., 2019; Johnson et al., 2017b) and Deep Sets (Zaheer et al., 2017), are effective on specific tasks.

However, there is limited understanding of the relation between the generalization ability and network structure for reasoning. *What tasks can a neural network (sample efficiently) learn to reason about?* Answering this question is crucial for understanding the empirical success and limitations of existing models, and for designing better models for new reasoning tasks.

This paper is an initial work towards answering this fundamental question, by developing a theoretical framework to characterize what tasks a neural network can reason about. We build on a simple observation that reasoning processes resemble algorithms. Hence, we study how well a reasoning algorithm *aligns* with the computation graph of the network. Intuitively, if they align well, the network only needs to learn *simple* algorithm steps to simulate the reasoning process, which leads to better sample efficiency. We formalize this intuition with a numeric measure of algorithmic alignment,

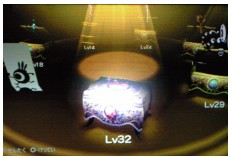 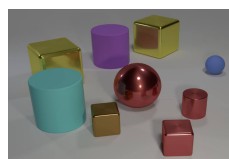 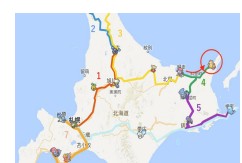 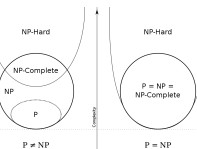

*Summary statistics*
What is the maximum value difference among treasures?

*Relational argmax*
What are the colors of the furthest pair of objects?

*Dynamic programming*
What is the cost to defeat monster X by following the optimal path?

*NP-hard problem*
Subset sum: Is there a subset that sums to 0?

Figure 1: **Overview of reasoning tasks with increasingly complex structure.** Each *task category* shows an example task on which we perform experiments in Section 4. Algorithmic alignment suggests that (a) Deep Sets and GNNs, but not MLP, can *sample efficiently* learn summary statistics, (b) GNNs, but not Deep Sets, can learn relational argmax, (c) GNNs can learn dynamic programming, an algorithmic paradigm that we show to unify many reasoning tasks, (d) GNNs cannot learn subset sum (NP-hard), but NES, a network we design based on exhaustive search, can generalize. Our theory agrees with empirical results (Fig. 3).

and show initial support for our hypothesis that algorithmic alignment facilitates learning: Under simplifying assumptions, we show a sample complexity bound that decreases with better alignment.

Our framework explains the empirical success of popular reasoning models and suggests their limitations. As concrete examples, we study four categories of increasingly complex reasoning tasks: summary statistics, relational argmax (asking about properties of the result of comparing multiple relations), dynamic programming, and NP-hard problems (Fig. 1). Using alignment, we characterize which architectures are expected to learn each task well: Networks inducing permutation invariance, such as Deep Sets (Zaheer et al., 2017), can learn summary statistics, and one-iteration GNNs can learn relational argmax. Many other more complex tasks, such as intuitive physics, visual question answering, and shortest paths – despite seeming different – can all be solved via a powerful algorithmic paradigm: dynamic programming (DP) (Bellman, 1966). Multi-iteration GNNs algorithmically align with DP and hence are expected to sample-efficiently learn these tasks. Indeed, they do. Our results offer an explanation for the popularity of GNNs in the relational reasoning literature, and also suggest limitations for tasks with even more complex structure. As an example of such a task, we consider subset sum, an NP-hard problem where GNNs indeed fail. Overall, empirical results (Fig. 3) agree with our theoretical analysis based on algorithmic alignment (Fig. 1). These findings also suggest how to take into account task structure when designing new architectures.

The perspective that structure in networks helps is not new. For example, in a well-known position paper, Battaglia et al. (2018) argue that GNNs are suitable for relational reasoning because they have relational inductive biases, but without formalizations. Here, we take such ideas one step further, by introducing a *formal* definition (algorithmic alignment) for quantifying the relation between network and task structure, and by formally deriving implications for learning. These theoretical ideas are the basis for characterizing what reasoning tasks a network can learn well. Our algorithmic structural condition also differs from structural assumptions common in learning theory (Vapnik, 2013; Bartlett & Mendelson, 2002; Bartlett et al., 2017; Neyshabur et al., 2015; Golowich et al., 2018) and specifically aligns with reasoning.

In summary, we introduce algorithmic alignment to analyze learning for reasoning. Our initial theoretical results suggest that algorithmic alignment is desirable for generalization. On four categories of reasoning tasks with increasingly complex structure, we apply our framework to analyze which tasks some popular networks can learn well. GNNs algorithmically align with dynamic programming, which solves a broad range of reasoning tasks. Finally, our framework implies guidelines for designing networks for new reasoning tasks. Experimental results confirm our theory.

## 2 Preliminaries

We begin by introducing notations and summarizing common neural networks for reasoning tasks. Let $S$ denote the universe, *i.e.*, a configuration/set of objects to reason about. Each object $s \in S$ is represented by a vector $X$. This vector could be state descriptions (Battaglia et al., 2016; Santoro

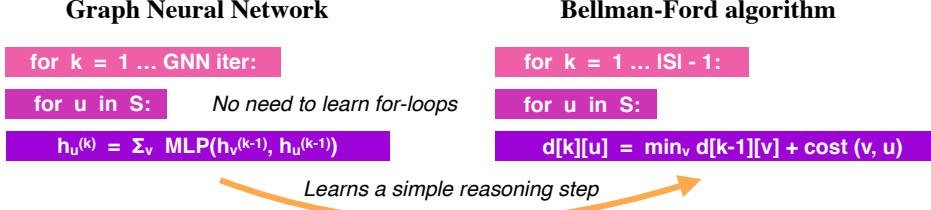

Figure 2: **Our framework suggests that better algorithmic alignment improves generalization.** As an example, our framework explains why GNN generalizes when learning to answer shortest paths. A correct reasoning process for the shortest paths task is the Bellman-Ford algorithm. The computation structure of a GNN (left) *aligns* well with Bellman-Ford (right): the GNN can simulate Bellman-Ford by merely learning a *simple reasoning step*, i.e., the relaxation step in the last line (a sum, and a min over neighboring nodes $v$) via its aggregation operation. In contrast, a giant MLP or Deep Set must learn the structure of the *entire for-loop*. Thus, the GNN is expected to generalize better when learning shortest paths, as is confirmed in experiments (Section 4.3).

et al., 2017) or features learned from data such as images (Santoro et al., 2017). Information about the specific question can also be included in the object representations. Given a set of universes $\{S_1, ..., S_M\}$ and answer labels $\{y_1, ..., y_M\} \subseteq \mathcal{Y}$, we aim to learn a function $g$ that can answer questions about unseen universes, $y = g(S)$.

**Multi-layer perceptron (MLP).** For a single-object universe, applying an MLP on the object representation usually works well. But when there are multiple objects, simply applying an MLP to the concatenated object representations often does not generalize (Santoro et al., 2017).

**Deep Sets.** As the input to the reasoning function is an unordered set, the function should be permutation-invariant, i.e., the output is the same for all input orderings. To induce permutation invariance in a neural network, Zaheer et al. (2017) propose *Deep Sets*, of the form

$$y = \mathrm{MLP}_2\Big(\sum_{s \in S} \mathrm{MLP}_1(X_s)\Big). \tag{2.1}$$

**Graph Neural Networks (GNNs).** GNNs are originally proposed for learning on graphs (Scarselli et al., 2009b). Their structures follow a message passing scheme (Gilmer et al., 2017; Xu et al., 2018; 2019), where the representation $h_s^{(k)}$ of each node $s$ (in iteration $k$) is recursively updated by aggregating the representation of neighboring nodes. GNNs can be adopted for reasoning by considering objects as nodes and assuming all objects pairs are connected, *i.e.*, a complete graph (Battaglia et al., 2018):

$$h_s^{(k)} = \sum_{t \in S} \mathrm{MLP}_1^{(k)}\left(h_s^{(k-1)}, h_t^{(k-1)}\right), \quad h_S = \mathrm{MLP}_2\Big(\sum_{s \in S} h_s^{(K)}\Big), \tag{2.2}$$

where $h_S$ is the answer/output and $K$ is the number of GNN layers. Each object's representation is initialized as $h_s^{(0)} = X_s$. Although other aggregation functions are proposed, we use sum in our experiments. Similar to Deep Sets, GNNs are also permutation invariant. While Deep Sets focus on individual objects, GNNs can also focus on pairwise relations.

The GNN framework includes many reasoning models. Relation Networks (Santoro et al., 2017) and Interaction Networks (Battaglia et al., 2016) resemble one-layer GNNs. Recurrent Relational Networks (Palm et al., 2018) apply LSTMs (Hochreiter & Schmidhuber, 1997) after aggregation.

## 3 THEORETICAL FRAMEWORK: ALGORITHMIC ALIGNMENT

Next, we study how the network structure and task may interact, and possible implications for generalization. Empirically, different network structures have different degrees of success in learning reasoning tasks, e.g., GNNs can learn relations well, but Deep Sets often fail (Fig. 3). However, all these networks are universal approximators (Propositions 3.1 and 3.2). Thus, their differences in test accuracy must come from generalization.

We observe that the answer to many reasoning tasks may be computed via a reasoning algorithm; we further illustrate the algorithms for some reasoning tasks in Section 4. Many neural networks can *represent* algorithms (Pérez et al., 2019). For example, Deep Sets can universally represent permutation-invariant set functions (Zaheer et al., 2017; Wagstaff et al., 2019). This also holds for GNNs and MLPs, as we show in Propositions 3.1 and 3.2 (our setting differs from Scarselli et al. (2009a) and Xu et al. (2019), who study functions on graphs):

**Proposition 3.1.** *Let $f : \mathbb{R}^{d \times N} \to \mathbb{R}$ be any continuous function over sets $S$ of bounded cardinality $|S| \leq N$. If $f$ is permutation-invariant to the elements in $S$, and the elements are in a compact set in $\mathbb{R}^d$, then $f$ can be approximated arbitrarily closely by a GNN (of any depth).*

**Proposition 3.2.** *For any GNN $\mathcal{N}$, there is an MLP that can represent all functions $\mathcal{N}$ can represent.*

But, empirically, not all network structures work well when *learning* these algorithms, i.e., they generalize differently. Intuitively, a network may generalize better if it can represent a function "more easily". We formalize this idea by *algorithmic alignment*, formally defined in Definition 3.4. Indeed, not only the reasoning process has an algorithmic structure: the neural network's architecture induces a computational structure on the function it computes. This corresponds to an algorithm that prescribes how the network combines computations from modules. Fig. 2 illustrates this idea for a GNN, where the modules are its MLPs applied to pairs of objects. In the shortest paths problem, the GNN matches the structure of the Bellman-Ford algorithm: to simulate the Bellman-Ford with a GNN, the GNN's MLP modules only need to learn a *simple* update equation (Fig. 2). In contrast, if we want to represent the Bellman-Ford algorithm with a single MLP, it needs to simulate an *entire for-loop*, which is much more complex than one update step. Therefore, we expect the GNN to have better sample complexity than MLP when learning to solve shortest path problems.

This perspective suggests that a neural network which better aligns with a correct reasoning process (algorithmic solution) can more easily learn a reasoning task than a neural network that does not align well. If we look more broadly at reasoning, there may also exist solutions which only solve a task approximately, or whose structure is obtuse. In this paper, we focus on reasoning tasks whose underlying reasoning process is exact and has clear algorithmic structure. We leave the study of approximation algorithms and unknown structures for future work.

## 3.1 FORMALIZATION OF ALGORITHMIC ALIGNMENT

We formalize the above intuition in a PAC learning framework (Valiant, 1984). PAC learnability formalizes *simplicity* as sample complexity, i.e., the number of samples needed to ensure low test error with high probability. It refers to a learning algorithm $\mathcal{A}$ that, given training samples $\{x_i, y_i\}_{i=1}^M$, outputs a function $f = \mathcal{A}(\{x_i, y_i\}_{i=1}^M)$. The learning algorithm here is the neural network and its training method, e.g., gradient descent. A function is simple if it has low sample complexity.

**Definition 3.3. (PAC learning and sample complexity).** Fix an error parameter $\epsilon > 0$ and failure probability $\delta \in (0, 1)$. Suppose $\{x_i, y_i\}_{i=1}^M$ are i.i.d. samples from some distribution $\mathcal{D}$, and the data satisfies $y_i = g(x_i)$ for some underlying function $g$. Let $f = \mathcal{A}(\{x_i, y_i\}_{i=1}^M)$ be the function generated by a learning algorithm $\mathcal{A}$. Then $g$ is $(M, \epsilon, \delta)$-*learnable* with $\mathcal{A}$ if

$$\mathbb{P}_{x \sim \mathcal{D}} \left[ \|f(x) - g(x)\| \leq \epsilon \right] \geq 1 - \delta. \tag{3.1}$$

The *sample complexity* $\mathcal{C}_{\mathcal{A}} (g, \epsilon, \delta)$ is the minimum $M$ so that $g$ is $(M, \epsilon, \delta)$-learnable with $\mathcal{A}$.

With the PAC learning framework, we define a numeric measure of algorithmic alignment (Definition 3.4), and under simplifying assumptions, we show that the sample complexity decreases with better algorithmic alignment (Theorem 3.6).

Formally, a neural network aligns with an algorithm if it can simulate the algorithm via a limited number of modules, and each module is simple, i.e., has low sample complexity.

**Definition 3.4. (Algorithmic alignment).** Let $g$ be a reasoning function and $\mathcal{N}$ a neural network with $n$ modules $\mathcal{N}_i$. The module functions $f_1, ..., f_n$ generate $g$ for $\mathcal{N}$ if, by replacing $\mathcal{N}_i$ with $f_i$, the network $\mathcal{N}$ simulates $g$. Then $\mathcal{N}$ $(M, \epsilon, \delta)$-algorithmically aligns with $g$ if (1) $f_1, ..., f_n$ generate $g$ and (2) there are learning algorithms $\mathcal{A}_i$ for the $\mathcal{N}_i$'s such that $n \cdot \max_i C_{\mathcal{A}_i}(f_i, \epsilon, \delta) \leq M$.

Good algorithmic alignment, i.e., small $M$, implies that all algorithm steps $f_i$ to simulate the algorithm $g$ are *easy to learn*. Therefore, the algorithm steps should not simulate complex programming constructs such as for-loops, whose sample complexity is large (Theorem 3.5).

Next, we show how to compute the algorithmic alignment value $M$. Algorithmic alignment resembles Kolmogorov complexity (Kolmogorov, 1998) for neural networks. Thus, it is generally non-trivial to obtain the optimal alignment between a neural network and an algorithm. However, one important difference to Kolmogorov complexity is that any algorithmic alignment that yields decent sample complexity is good enough (unless we want the tightest bound). In Section 4, we will see several examples where finding a good alignment is not hard. Then, we can compute the value of an alignment by summing the sample complexity of the algorithm steps with respect to the modules, e.g. MLPs. For ilustration, we show an example of how one may compute sample complexity of MLP modules.

A line of works show one can analyze the optimization and generalization behavior of overparameterized neural networks via neural tangent kernel (NTK) (Allen-Zhu et al., 2019; Arora et al., 2019a;b; 2020; Du et al., 2019c;a; Jacot et al., 2018; Li & Liang, 2018). Building upon Arora et al. (2019a), Du et al. (2019b) show that infinitely-wide GNNs trained with gradient descent can provably learn certain smooth functions. The current work studies a broader class of functions, e.g., algorithms, compared to those studied in Du et al. (2019b), but with more simplifying assumptions.

Here, Theorem 3.5, proved in the Appendix, summarizes and extends Theorem 6.1 of Arora et al. (2019a) for over-parameterized MLP modules to vector-valued functions. Our framework can be used with other sample complexity bounds for other types of modules, too.

**Theorem 3.5.** *(Sample complexity for overparameterized MLP modules). Let $\mathcal{A}$ be an overparameterized and randomly initialized two-layer MLP trained with gradient descent for a sufficient number of iterations. Suppose $g : \mathbb{R}^d \to \mathbb{R}^m$ with components $g(x)^{(i)} = \sum_j \alpha_j^{(i)} \big(\beta_j^{(i)\top} x\big)^{p_j^{(i)}}$, where $\beta_j^{(i)} \in \mathbb{R}^d$, $\alpha \in \mathbb{R}$, and $p_j^{(i)} = 1$ or $p_j^{(i)} = 2l \, (l \in \mathbb{N}_+)$. The sample complexity $\mathcal{C}_{\mathcal{A}}(g, \epsilon, \delta)$ is*

$$\mathcal{C}_{\mathcal{A}}(g, \epsilon, \delta) = O\Big(\frac{\max_i \sum_{j=1}^K p_j^{(i)} |\alpha_j^{(i)}| \cdot \|\beta_j^{(i)}\|_2^{p_j^{(i)}} + \log\left(m/\delta\right)}{(\epsilon/m)^2}\Big). \tag{3.2}$$

Theorem 3.5 suggests that functions that are "simple" when expressed as a polynomial, e.g., via a Taylor expansion, are sample efficiently learnable by an MLP module. Thus, algorithm steps that perform computation over many objects may require many samples for an MLP module to learn, since the number $K$ of polynomials or $\|\beta_j^{(i)}\|$ can increase in Eqn. (3.2). "For loop" is one example of such complex algorithm steps.

## 3.2 BETTER ALGORITHMIC ALIGNMENT IMPLIES BETTER GENERALIZATION

We show an initial result demonstrating that algorithmic alignment is desirable for generalization. Theorem 3.6 states that, in a simplifying setting where we sequentially train modules of a network with auxiliary labels, the sample complexity bound increases with algorithmic alignment value $M$.

While we do not have auxiliary labels in practice, we observe the same pattern for end-to-end learning in experiments (Section 4). We leave sample complexity analysis for end-to-end-learning to future work. We prove Theorem 3.6 in Appendix D.

**Theorem 3.6.** *(Algorithmic alignment improves sample complexity). Fix $\epsilon$ and $\delta$. Suppose $\{S_i, y_i\}_{i=1}^M \sim \mathcal{D}$, where $|S_i| < N$, and $y_i = g(S_i)$ for some $g$. Suppose $\mathcal{N}_1, ..., \mathcal{N}_n$ are network $\mathcal{N}$'s MLP modules in sequential order. Suppose $\mathcal{N}$ and $g$ $(M, \epsilon, \delta)$-algorithmically align via functions $f_1, ..., f_n$. Under the following assumptions, $g$ is $(M, O(\epsilon), O(\delta))$-learnable by $\mathcal{N}$.*

*a) Algorithm stability. Let $\mathcal{A}$ be the learning algorithm for the $\mathcal{N}_i$'s. Suppose $f = \mathcal{A}(\{x_i, y_i\}_{i=1}^M)$, and $\hat{f} = \mathcal{A}(\{\hat{x}_i, y_i\}_{i=1}^M)$. For any $x$, $\|f(x) - \hat{f}(x)\| \leq L_0 \cdot \max_i \|x_i - \hat{x}_i\|$, for some $L_0$.*

*b) Sequential learning. We train $\mathcal{N}_i$'s sequentially: $\mathcal{N}_1$ has input samples $\{\hat{x}_i^{(1)}, f_1(\hat{x}_i^{(1)})\}_{i=1}^N$, with $\hat{x}_i^{(1)}$ obtained from $S_i$. For $j > 1$, the input $\hat{x}_i^{(j)}$ for $\mathcal{N}_j$ are the outputs from the previous modules, but labels are generated by the correct functions $f_{j-1}, ..., f_1$ on $\hat{x}_i^{(1)}$.*

*c) Lipschitzness. The learned functions $\hat{f}_j$ satisfy $\|\hat{f}_j(x) - \hat{f}_j(\hat{x})\| \leq L_1 \|x - \hat{x}\|$, for some $L_1$.*

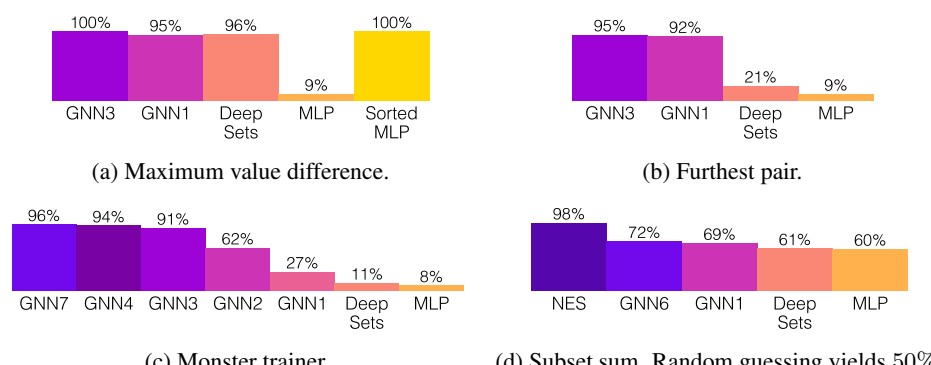

(a) Maximum value difference.

(b) Furthest pair.

(c) Monster trainer.

(d) Subset sum. Random guessing yields $50\%$.

Figure 3: **Test accuracies on reasoning tasks with increasingly complex structure.** Fig. 1 shows an overview of the tasks. GNN$k$ is GNN with $k$ iterations. (a) Summary statistics. All models except MLP generalize. (b) Relational argmax. Deep Sets fail. (c) Dynamic programming. Only GNNs with sufficient iterations generalize. (d) An NP-hard problem. Even GNNs fail, but NES generalizes.

In our analysis, the Lipschitz constants and the universe size are constants going into $O(\epsilon)$ and $O(\delta)$. As an illustrative example, we use Theorem 3.6 and 3.5 to show that GNN has a polynomial improvement in sample complexity over MLP when learning simple relations. Indeed, GNN aligns better with summary statistics of pairwise relations than MLP does (Section 4.1).

**Corollary 3.7.** *Suppose universe $S$ has $\ell$ objects $X_1, ..., X_\ell$, and $g(S) = \sum_{i,j}(X_i - X_j)^2$. In the setting of Theorem 3.6, the sample complexity bound for MLP is $O(\ell^2)$ times larger than for GNN.*

## 4 PREDICTING WHAT NEURAL NETWORKS CAN REASON ABOUT

Next, we apply our framework to analyze the neural networks for reasoning from Section 2: MLP, Deep Sets, and GNNs. Using algorithmic alignment, we predict whether each model can generalize on four categories of increasingly complex reasoning tasks: summary statistics, relational argmax, dynamic programming, and an NP-hard problem (Fig. 3). Our theoretical analysis is confirmed with experiments (Dataset and training details are in Appendix G). To empirically compare sample complexity of different models, we make sure all models perfectly fit training sets through extensive hyperparameter tuning. Therefore, the test accuracy reflects how well a model generalizes.

The examples in this section, together with our framework, suggest an explanation why GNNs are widely successful across reasoning tasks: Popular reasoning tasks such as visual question answering and intuitive physics can be solved by DP. GNNs align well with DP, and hence are expected to learn sample efficiently.

### 4.1 SUMMARY STATISTICS

As discussed in Section 2, we assume each object $X$ has a state representation $X = [h_1, h_2, ..., h_k]$, where each $h_i \in \mathbb{R}^{d_i}$ is a feature vector. An MLP can learn simple polynomial functions of the state representation (Theorem 3.5). In this section, we show how Deep Sets use MLP as building blocks to learn summary statistics.

Questions about summary statistics are common in reasoning tasks. One example from CLEVR (Johnson et al., 2017a) is "How many objects are either small cylinders or red things?" Deep Sets (Eqn. 2.1) align well with algorithms that compute summary statistics over individual objects. Suppose we want to compute the sum of a feature over all objects. To simulate the reasoning algorithm, we can use the first MLP in Deep Sets to extract the desired feature and aggregate them using the pooling layer. Under this alignment, each MLP only needs to learn simple steps, which leads to good sample complexity. Similarly, Deep Sets can learn to compute *max or min* of a feature by using smooth approximations like the softmax $\max_{s \in S} X_s \approx \log(\sum_{s \in X_s} \exp(X_s))$. In contrast, if we train an MLP to perform sum or max, the MLP must learn a complex for-loop and therefore needs more

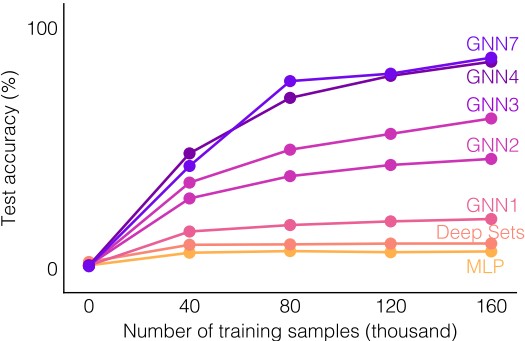

Figure 4: **Test accuracy vs. training set size** for models trained on sub-sampled training sets and evaluated on the same test set of monster trainer (DP task). Test accuracies increase faster when a neural network aligns well with an algorithmic solution of the task. For example, the test accuracy of GNN4 increases by $23\%$ when the number of training samples increases from $40,000$ to $80,000$, which is much higher than that of Deep Sets ($0.2\%$).

samples. Therefore, our framework predicts that Deep Sets have better sample complexity than MLP when learning summary statistics.

**Maximum value difference.** We confirm our predictions by training models to compute the *maximum value difference* task. Each object in this task is a treasure $X = [h_1, h_2, h_3]$ with location $h_1$, value $h_2$, and color $h_3$. We train models to predict the difference in value between the most and the least valuable treasure, $y(S) = \max_{s \in S} h_2(X_s) - \min_{s \in S} h_2(X_s)$.

The test accuracy follows our prediction (Fig. 3a). MLP does not generalize and only has 9% test accuracy, while Deep Sets has 96%. Interestingly, if we sort the treasures by value (Sorted MLP in Fig. 3a), MLP achieves perfect test accuracy. This observation can be explained with our theory—when the treasures are sorted, the reasoning algorithm is reduced to a simple subtraction: $y(S) = h_2(X_{|S|}) - h_2(X_1)$, which has a low sample complexity for even MLPs (Theorem 3.5). GNNs also have high test accuracies. This is because summary statistics are a special case of relational argmax, which GNNs can learn as shown next.

## 4.2 RELATIONAL ARGMAX

Next, we study *relational argmax*: tasks where we need to compare pairwise relations and answer a question about that result. For example, a question from Sort-of-CLEVR (Santoro et al., 2017) asks "What is the shape of the object that is farthest from the gray object?", which requires comparing the distance between object pairs.

One-iteration GNN aligns well with relational argmax, as it sums over all pairs of objects, and thus can compare, e.g. via softmax, pairwise information without learning the "for loops". In contrast, Deep Sets require many samples to learn this, because most pairwise relations cannot be encoded as a sum of individual objects:

**Claim 4.1.** *Suppose $g(x, y) = 0$ if and only if $x = y$. There is no $f$ such that $g(x, y) = f(x) + f(y)$.*

Therefore, if we train a Deep Set to compare pairwise relations, one of the MLP modules has to learn a complex "for loop", which leads to poor sample complexity. Our experiment confirms that GNNs generalize better than Deep Sets when learning relational argmax.

**Furthest pair.** As an example of relational argmax, we train models to identify the furthest pair among a set of objects. We use the same object settings as the maximum value difference task. We train models to find the colors of the two treasures with the largest distance. The answer is a pair of colors, encoded as an integer category:

$$y(S) = (h_3(X_{s_1}), h_3(X_{s_2})) \quad \text{s.t.} \; \{X_{s_1}, X_{s_2}\} = \arg\max_{s_1, s_2 \in S} \|h_1(X_{s_1}) - h_1(X_{s_2})\|_{\ell_1}$$

Distance as a pairwise function satisfies the condition in Claim 4.1. As predicted by our framework, Deep Sets has only 21% test accuracy, while GNNs have more than 90% accuracy.

## 4.3 DYNAMIC PROGRAMMING

We observe that a broad class of relational reasoning tasks can be unified by the powerful algorithmic paradigm *dynamic programming (DP)* (Bellman, 1966). DP recursively breaks down a problem into simpler sub-problems. It has the following general form:

$$\text{Answer}[k][i] = \text{DP-Update}(\{\text{Answer}[k-1][j]\}, j = 1...n), \quad (4.1)$$

where $\text{Answer}[k][i]$ is the solution to the sub-problem indexed by iteration $k$ and state $i$, and DP-Update is an task-specific update function that computes $\text{Answer}[k][i]$ from $\text{Answer}[k-1][j]$'s.

GNNs algorithmically align with a class of DP algorithms. We can interpret GNN as a DP algorithm, where node representations $h_i^{(k)}$ are $\text{Answer}[k][i]$, and the GNN aggregation step is the DP-Update. Therefore, Theorem 3.6 suggests that a GNN with enough iterations can sample efficiently learn any DP algorithm with a simple DP-update function, e.g. sum/min/max.

**Shortest paths.** As an example, we experiment with GNN on Shortest paths, a standard DP problem. Shortest paths can be solved by the Bellman-Ford algorithm (Bellman, 1958), which recursively updates the minimum distance between each object $u$ and the source $s$:

$$\text{distance}[1][u] = \text{cost}(s, u), \quad \text{distance}[k][u] = \min_v \{\text{distance}[k-1][v] + \text{cost}(v, u)\}, \quad (4.2)$$

As discussed above, GNN aligns well with this DP algorithm. Therefore, our framework predicts that GNN has good sample complexity when learning to find shortest paths. To verify this, we test different models on a **monster trainer** game, which is a shortest path variant with unkown cost functions that need to be learned by the models. Appendix G.3 describes the task in details.

In Fig. 3c, only GNNs with at least four iterations generalize well. The empirical result confirms our theory: a neural network can sample efficiently learn a task if it aligns with a correct algorithm. Interestingly, GNN does not need as many iterations as Bellman-Ford. While Bellman-Ford needs $N = 7$ iterations, GNNs with four iterations have almost identical test accuracy as GNNs with seven iterations (94% vs 95%). This can also be explained through algorithmic alignment, as GNN aligns with an optimized version of Bellman-Ford, which we explain in Appendix G.3.

Fig. 4 shows how the test accuracies of different models vary with the number of sub-sampled training points. Indeed, the test accuracy increases more slowly for models that align worse with the task, which implies they need more training samples to achieve similar generalization performance. Again, this confirms our theory.

After verifying that GNNs can sample-efficiently learn DP, we show that two popular families of reasoning tasks, visual question answering and intuitive physics, can be formulated as DP. Therefore, our framework explains why GNNs are effective in these tasks.

**Visual question answering.** The Pretty-CLEVR dataset (Palm et al., 2018) is an extension of Sort-of-CLEVR (Santoro et al., 2017) and CLEVR (Johnson et al., 2017a). GNNs work well on these datasets. Each question in Pretty-CLEVR has state representations and asks "Starting at object $X$, if each time we jump to the closest object, which object is $K$ jumps away?". This problem can be solved by DP, which computes the answers for $k$ jumps from the answers for $(k-1)$ jumps.

$$\text{closest}[1][i] = \arg\min_j d(i, j), \quad \text{closest}[k][i] = \text{closest}[k-1]\Big[\text{closest}[1][i]\Big] \text{ for } k > 1, \quad (4.3)$$

where $\text{closest}[k][i]$ is the answer for jumping $k$ times from object $i$, and $d(i, j)$ is the distance between the $i$-th and the $j$-th object.

**Intuitive physics.** Battaglia et al. (2016) and Watters et al. (2017) train neural networks to predict object dynamics in rigid body scenes and n-body systems. Chang et al. (2017) and Janner et al. (2019) study other rigid body scenes. If the force acting on a physical object stays constant, we can compute the object's trajectory with simple functions (physics laws) based on its initial position and force. Physical interactions, however, make the force change, which means the function to compute the object's dynamics has to change too. Thus, a DP algorithm would *recursively* compute the next force changes in the system and update DP states (velocity, momentum, position etc of objects) according to the (learned) forces and physics laws (Thijssen, 2007).

$$\text{for } k = 1..K : \quad \text{time} = \min_{i,j} \text{Force-change-time}(\text{state}[k-1, i], \text{state}[k-1, j]), \quad (4.4)$$

$$\text{for } i = 1..N : \quad \text{state}[k][i] = \text{Update-by-forces}(\text{state}[k-1][j], \text{time}), \ j = 1..N, \quad (4.5)$$

Force-change-time computes the time at which the force between object $i$ and $j$ will change. Update-by-forces updates the state of each object at the next force change time. In rigid body systems, force changes only at collision. In datasets where no object collides more than once between time frames, one-iteration algorithm/GNN can work (Battaglia et al., 2016). More iterations are needed if multiple collisions occur between two consecutive frames (Li & Liang, 2018). In n-body systems, forces change continuously but smoothly. Thus, finite-iteration DP/GNN can be viewed as a form of Runge-Kutta method (DeVries & Hamill, 1995).

## 4.4 DESIGNING NEURAL NETWORKS WITH ALGORITHMIC ALIGNMENT

While DP solves many reasoning tasks, it has limitations. For example, NP-hard problems cannot be solved by DP. It follows that GNN also cannot sample-efficiently learn these hard problems. Our framework, however, goes beyond GNNs. If we know the structure of a suitable underlying reasoning algorithm, we can design a network with a similar structure to learn it. If we have no prior knowledge about the structure, then neural architecture search over algorithmic structures will be needed.

**Subset Sum.** As an example, we design a new architecture that can learn to solve the subset sum problem: Given a set of numbers, does there exist a subset that sums to $0$? Subset sum is NP-hard (Karp, 1972) and cannot be solved by DP. Therefore, our framework predicts that GNN cannot generalize on this task. One subset sum algorithm is exhaustive search, where we enumerate all $2^{|S|}$ possible subsets $\tau$ and check whether $\tau$ has zero-sum. Following this algorithm, we design a similarly structured neural network which we call **Neural Exhaustive Search (NES)**. Given a universe, NES enumerates all subsets of objects and passes each subset through an LSTM followed by a MLP. The results are aggregated with a max-pooling layer and MLP:

$$\text{MLP}_2(\max_{\tau \subseteq S} \text{MLP}_1 \circ \text{LSTM}(X_1, ..., X_{|\tau|} : X_1, ..., X_{|\tau|} \in \tau)). \tag{4.6}$$

This architecture aligns well with subset-sum, since the first MLP and LSTM only need to learn a simple step, checking whether a subset has zero sum. Therefore, we expect NES to generalize well in this task. Indeed, NES has 98% test accuracy, while other models perform much worse (Fig. 3d).

## 5 CONCLUSION

This paper is an initial step towards formally understanding how neural networks can learn to reason. In particular, we answer what tasks a neural network can learn to reason about well, by studying the generalization ability of learning the underlying reasoning processes for a task. To this end, we introduce an algorithmic alignment framework to formalize the interaction between the structure of a neural network and a reasoning process, and provide preliminary results on sample complexity. Our results explain the success and suggest the limits of current neural architectures: Graph Neural Networks generalize in many popular reasoning tasks because the underlying reasoning processes for those tasks resemble dynamic programming.

Our algorithmic alignment perspective may inspire neural network design and opens up theoretical avenues. An interesting direction for future work is to design, e.g. via algorithmic alignment, neural networks that can learn other reasoning paradigms beyond dynamic programming, and to explore the neural architecture search space of algorithmic structures.

From a broader standpoint, reasoning assumes a good representation of the concepts and objects in the world. To complete the picture, it would also be interesting to understand how to better disentangle and eventually integrate "representation" and "reasoning".

### ACKNOWLEDGMENTS

We thank Zi Wang and Jiajun Wu for insightful discussions. This research was supported by NSF CAREER award 1553284, DARPA DSO's Lagrange program under grant FA86501827838 and a Chevron-MIT Energy Fellowship. This research was also supported by JST ERATO JPMJER1201 and JSPS Kakenhi JP18H05291. MZ was supported by DARPA award HR0011-15-C-0113 under subcontract to Raytheon BBN Technologies. The views, opinions, and/or findings contained in this article are those of the author and should not be interpreted as representing the official views or policies, either expressed or implied, of the Defense Advanced Research Projects Agency or the Department of Defense.

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

# A    PROOF OF PROPOSITION 3.1

We will prove the universal approximation of GNNs by showing that GNNs have at least the same expressive power as Deep Sets, and then apply the universal approximation of Deep Sets for permutation invariant continuous functions.

Zaheer et al. (2017) prove the universal approximation of Deep Sets under the restriction that the set size is fixed and the hidden dimension is equal to the set size plus one. Wagstaff et al. (2019) extend the universal approximation result for Deep Sets by showing that the set size does not have to be fixed and the hidden dimension is only required to be at least as large as the set size. The results for our purposes can be summarized as follows.

**Universal approximation of Deep Sets.**  Assume the elements are from a compact set in $\mathbb{R}^d$. Any continuous function on a set $S$ of size bounded by $N$, *i.e.*, $f : \mathbb{R}^{d \times N} \to \mathbb{R}$, that is permutation invariant to the elements in $S$ can be approximated arbitrarily close by some Deep Sets model with sufficiently large width and output dimension for its MLPs.

Next we show any Deep Sets can be expressed by some GNN with one message passing iteration. The computation structure of one-layer GNNs is shown below.

$$h_s = \sum_{t \in S} \phi\left(X_s, X_t\right), \quad h_S = g\left(\sum_{s \in S} h_s\right), \tag{A.1}$$

where $\phi$ and $g$ are parameterized by MLPs. If $\phi$ is a function that ignores $X_t$ so that $\phi\left(X_s, X_t\right) = \rho(X_s)$ for some $\rho$, *e.g.*, by letting part of the weight matricies in $\phi$ be 0, then we essentially get a Deep Sets in the following form.

$$h_s = \rho\left(X_s\right), \quad h_S = g\left(\sum_{s \in S} h_s\right). \tag{A.2}$$

For any such $\rho$, we can get the corresponding $\phi$ via the construction above. Hence for any Deep Sets, we can express it with an one-layer GNN. The same result applies to GNNs with multiple layers (message passing iterations), because we can express a function $\rho(X_s)$ by the composition of multiple $\rho^{(k)}$'s, which we can express with a GNN layer via our construction above. It then follows that GNNs are universal approximators for permutation invariant continuous functions.

# B    PROOF OF PROPOSITION 3.2

For any GNN $\mathcal{N}$, we construct an MLP that is able to do the exact same computation as $\mathcal{N}$. It will then follow that the MLP can represent any function $\mathcal{N}$ can represent. Suppose the computation structure of $\mathcal{N}$ is the following.

$$h_s^{(k)} = \sum_{t \in S} f^{(k)}\left(h_s^{(k-1)}, h_t^{(k-1)}\right), \quad h_S = g\left(\sum_{s \in S} h_s^{(K)}\right), \tag{B.1}$$

where $f$ and $g$ are parameterized by MLPs. Suppose the set size is bounded by $M$ (the expressive power of GNNs also depend on $M$ Wagstaff et al. (2019)). We first show the result for a fixed size input, *i.e.*, MLPs can simulate GNNs if the input set has a fixed size, and then apply an ensemble approach to deal with variable sized input.

Let the input to the MLP be a vector concatenated by $h_s^{(0)}$'s, in some arbitrary ordering. For each message passing iteration of $\mathcal{N}$, any $f^{(k)}$ can be represented by an MLP. Thus, for each pair of $(h_t^{(k-1)}, h_s^{(k-1)})$, we can set weights in the MLP so that the the concatenation of all $f(h_t^{(k-1)}, h_s^{(k-1)})$ become the hidden vector after some layers of the MLP. With the vector of $f(h_t^{(k-1)}, h_s^{(k-1)})$ as input, in the next few layers of the MLP we can construct weights so that we have the concatenation of $h_s^{(k)} = \sum_{t \in S} f^{(k)}\left(h_s^{(k-1)}, h_t^{(k-1)}\right)$ as the result of the hidden dimension, because we can encode summation with weights in MLPs. So far, we can simulate an iteration of GNN $\mathcal{N}$ with layers of MLP. We can repeat the process for $K$ times by stacking the similar layers. Finally, with a concatenation of

$h_s^{(K)}$ as our hidden dimension in the MLP, similarly, we can simulate $h_S = g\left(\sum_{s \in S} h_s^{(K)}\right)$ with layers of MLP. Stacking all layers together, we have obtained an MLP that can simulate $\mathcal{N}$.

To deal with variable sized inputs, we construct $M$ MLPs that can simulate the GNN for each input set size $1, ..., M$. Then we construct a meta-layer, whose weights represent (universally approximate) the summation of the output of $M$ MLPs multiplied by an indicator function of whether each MLPs has the same size as the set input (these need to be input information). The meta layer weights on top can then essentially select the output from of MLP that has the same size as the set input and then exactly simulate the GNN. Note that the MLP we construct here has the requirement for how we input the data and the information of set sizes etc. In practice, we can have $M$ MLPs and decide which MLP to use depending on the input set size.

## C    PROOF OF THEOREM 3.5

Theorem 3.5 is a generalization of Theorem 6.1 in (Arora et al., 2019a), which addresses the scalar case. See (Arora et al., 2019a) for a complete list of assumptions.

**Theorem C.1.** *(Arora et al., 2019a) Suppose we have $g : \mathbb{R}^d \to \mathbb{R}$, $g(x) = \sum_j \alpha_j \left(\beta_j^\top x\right)^{p_j}$, where $\beta_j \in \mathbb{R}^d$, $\alpha \in \mathbb{R}$, and $p_j = 1$ or $p_j = 2l$ $(l \in \mathbb{N}_+)$. Let $\mathcal{A}$ be an overparameterized two-layer MLP that is randomly initialized and trained with gradient descent for a sufficient number of iterations. The sample complexity $\mathcal{C}_{\mathcal{A}}(g, \epsilon, \delta)$ is $O\left(\frac{\sum_j p_j |\alpha_j| \cdot \|\beta_j\|_2^{p_j} + \log(1/\delta)}{\epsilon^2}\right)$.*

To extend the sample complexity bound to vector-valued functions, we view each entry/component of the output vector as an independent scalar-valued output. We can then apply a union bound to bound the error rate and failure probability for the output vector, and thus, bound the overall sample complexity.

Let $\epsilon$ and $\delta$ be the given error rate and failure probability. Moreover, suppose we choose some error rate $\epsilon_0$ and failure probability $\delta_0$ for the output/function of each entry. Applying Theorem C.1 to each component

$$g(x)^{(i)} = \sum_j \alpha_j^{(i)} \left(\beta_j^{(i)\top} x\right)^{p_j^{(i)}} =: g_i(x) \tag{C.1}$$

yields a sample complexity bound of

$$\mathcal{C}_{\mathcal{A}}(g_i, \epsilon_0, \delta_0) = O\left(\frac{\sum_j p_j^{(i)} |\alpha_j^{(i)}| \cdot \|\beta_j^{(i)}\|_2^{p_j^{(i)}} + \log(1/\delta_0)}{\epsilon_0^2}\right) \tag{C.2}$$

for each $g_i(x)$. Now let us bound the overall error rate and failure probability given $\epsilon_0$ and $\delta_0$ for each entry. The probability that we fail to learn each of the $g_i$ is at most $\delta_0$. Hence, by a union bound, the probability that we fail to learn any of the $g_i$ is at most $m \cdot \delta_0$. Thus, with probability at least $1 - m\delta_0$, we successfully learn all $g_i$ for $i = 1, ..., m$, so the error for every entry is bounded by $\epsilon_0$. The error for the vector output is then at most $\sum_{i=1}^m \epsilon_0 = m\epsilon_0$.

Setting $m\delta_0 = \delta$ and $m\epsilon_0 = \epsilon$ gives us $\delta_0 = \frac{\delta}{m}$ and $\epsilon_0 = \frac{\epsilon}{m}$. Thus, if we can successfully learn the function for each output entry independently with error $\epsilon/m$ and failure rate $\delta/m$, we can successfully learn the entire vector-valued function with rate $\epsilon$ and $\delta$. This yields the following overall sample complexity bound:

$$\mathcal{C}_{\mathcal{A}}(g, \epsilon, \delta) = O\left(\frac{\max_i \sum_j p_j^{(i)} |\alpha_j^{(i)}| \cdot \|\beta_j^{(i)}\|_2^{p_j^{(i)}} + \log(m/\delta)}{(\epsilon/m)^2}\right) \tag{C.3}$$

Regarding $m$ as a constant, we can further simplify the sample complexity to

$$\mathcal{C}_{\mathcal{A}}(g, \epsilon, \delta) = O\left(\frac{\max_i \sum_j p_j^{(i)} |\alpha_j^{(i)}| \cdot \|\beta_j^{(i)}\|_2^{p_j^{(i)}} + \log(1/\delta)}{\epsilon^2}\right). \tag{C.4}$$

## D    PROOF OF THEOREM 3.6

We will show the learnability result by an inductive argument. Specifically, we will show that under our setting and assumptions, the error between the learned function and correct function on the test set will not blow up after the transform of another learned function $\hat{f}_j$, assuming learnability on previous $\hat{f}_1, ..., \hat{f}_{j-1}$ by induction. Thus, we can essentially provably learn at all layers/iterations and eventually learn $g$.

Suppose we have performed the sequential learning. Let us consider what happens at the test time. Let $f_j$ be the *correct functions* as defined in the algorithmic alignment. Let $\hat{f}_j$ be the functions learned by algorithm $\mathcal{A}_j$ and MLP $\mathcal{N}_j$. We have input $S \sim \mathcal{D}$, and our goal is to bound $\|g(S) - \hat{g}(S)\|$ with high probability. To show this, we bound the error of the intermediate representation vectors, *i.e.*, the output of $\hat{f}_j$ and $f_j$, and thus, the input to $\hat{f}_{j+1}$ and $f_{j+1}$.

Let us first consider what happens for the first module $\mathcal{N}_1$. $f_1$ and $\hat{f}_1$ have the same input distribution $x \sim \mathcal{D}$, where $x$ are obtained from $S$, *e.g.*, the pairwise object representations as in Eqn. 2.2. Hence, by the learnability assumption on $\mathcal{A}_1$, $\|f_1(x) - \hat{f}_1(x)\| < \epsilon$ with probability at least $1 - \delta$. The error for the input of $\mathcal{N}_2$ is then $O(\epsilon)$ with failure probability $O(\delta)$, because there are a constant number of terms of aggregation of $f_1$'s output, and we can apply union bound to upper bound the failure probability.

Next, we proceed by induction. Let us fix a $k$. Let $z$ denote the input for $f_k$, which are generated by the previous $f_j$'s, and let $\hat{z}$ denote the input for $\hat{f}_k$, which are generated by the previous $\hat{f}_j$'s. Assume $\|z - \hat{z}\| \le O(\epsilon)$ with failure probability at most $O(\delta)$. We aim to show that this holds for $k + 1$. For the simplicity of notation, let $f$ denote the correct function $f_k$ and let $\hat{f}$ denote the learned function $\hat{f}_k$. Since there are a constant number of terms for aggregation, our goal is then to bound $\|\hat{f}(\hat{z}) - f(z)\|$. By triangle inequality, we have

$$\|\hat{f}(\hat{z}) - f(z)\| = \|\hat{f}(\hat{z}) - \hat{f}(z) + \hat{f}(z) - f(z)\| \tag{D.1}$$

$$\le \|\hat{f}(\hat{z}) - \hat{f}(z)\| + \|\hat{f}(z) - f(z)\| \tag{D.2}$$

We can bound the first term with the Lipschitzness assumption of $\hat{f}$ as the following.

$$\|\hat{f}(\hat{z}) - \hat{f}(z)\| \le L_1 \|\hat{z} - z\| \tag{D.3}$$

To bound the second term, our key insight is that $f$ is a *learnale correct function*, so by the learnability coefficients in algorithmic alignment, it is close to the function $\tilde{f}$ learned by the learning algorithm $\mathcal{A}$ on the *correct samples*, *i.e.*, $f$ is close to $\tilde{f} = \mathcal{A}\left(\{z_i, y_i\}\right)$. Moreover, $\hat{f}$ is generated by the learning algorithm $\mathcal{A}$ on the perturbed samples, *i.e.*, $\hat{f} = \mathcal{A}\left(\{\hat{z}_i, y_i\}\right)$. By the algorithm stability assumption, $\hat{f}$ and $\tilde{f}$ should be close if the input samples are only slightly perturbed. It then follows that

$$\|\hat{f}(z) - f(z)\| = \|\hat{f}(z) - \tilde{f}(z) + \tilde{f}(z) - f(z)\| \tag{D.4}$$

$$\le \|\hat{f}(z) - \tilde{f}(z)\| + \|\tilde{f}(z) - f(z)\| \tag{D.5}$$

$$\le L_0 \max_i \|z_i - \hat{z}_i\| + \epsilon \quad \text{w.p. } \ge 1 - \delta \tag{D.6}$$

where $z_i$ and $\hat{z}_i$ are the training samples at the same layer $k$. Here, we apply the same induction condition as what we had for $z$ and $\hat{z}$: $\|z_i - \hat{z}_i\| \le O(\epsilon)$ with failure probability at most $O(\delta)$. We can then apply union bound to bound the probability of any bad event happening. Here, we have 3 bad events each happening with probability at most $O(\delta)$. Thus, with probability at least $1 - O(\delta)$, we have

$$\|\hat{f}(\hat{z}) - f(z)\| \le L_1 O(\epsilon) + L_0 O(\epsilon) + \epsilon = O(\epsilon) \tag{D.7}$$

This completes the proof.

## E    PROOF OF COROLLARY 3.7

Our main insight is that a giant MLP learns the same function $(X_i - X_j)^2$ for $\ell^2$ times and encode them in the weights. This leads to the $O(\ell^2)$ extra sample complexity through Theorem 3.5, because the number of polynomial terms $(X_i - X_j)^2$ is of order $\ell^2$.

First of all, the function $f(x, y) = (x - y)^2$ can be expressed as the following polynomial.

$$(x - y)^2 = \left( [1 \ -1]^\top [x \ y] \right)^2 \tag{E.1}$$

We have $\beta = [1 \ -1]$, so $p \cdot \|\beta\|^p = 4$. Hence, by Theorem 3.5, it takes $O(\frac{\log(1/\delta)}{\epsilon^2})$ samples for an MLP to learn $f(x, y) = (x - y)^2$. Under the sequential training setting, an one-layer GNN applies an MLP to learn $f$, and then sums up the outcome of $f(X_i, X_j)$ for all pairs $X_i, X_j$. Here, we essentially get the aggregation error $O(\ell^2 \cdot \epsilon)$ from $\ell^2$ pairs. However, we will see that applying an MLP to learn $g$ will also incur the same aggregation error. Hence, we do not need to consider the aggregation error effect when we compare the sample complexities.

Now we consider using MLP to learn the function $g$. No matter in what order the objects $X_i$ are concatenated, we can express $g$ with the sum of polynomials as the following.

$$g(S) = \sum_{ij} (\beta_{ij}^\top [X_1, ..., X_n])^2, \tag{E.2}$$

where $\beta_{ij}$ has 1 at the $i$-th entry, $-1$ at the $j$-th entry and 0 elsewhere. Hence $\|\beta_{ij}\|^p \cdot p = 4$. It then follows from Theorem 3.5 and union bound that it takes $O((\ell^2 + \log(1/\hat{\delta}))/\hat{\epsilon}^2)$ to learn $g$, where $\hat{\epsilon} = \ell^2 \epsilon$ and $\hat{\delta} = \ell^2 \delta$. Here, as we have discussed above, the same aggregation error $\hat{\epsilon}$ occurs in the aggregation process of $f$, so we can simply consider $\hat{\epsilon}$ for both. Thus, comparing $O(\log(1/\hat{\delta})/\hat{\epsilon}^2)$ and $O((\ell^2 + \log(1/\hat{\delta}))/\hat{\epsilon}^2)$ gives us the $O(\ell^2)$ difference.

## F  Proof of Claim 4.1

We prove the claim by contradiction. Suppose there exists $f$ such that $f(x) + f(y) = g(x, y)$ for any $x$ and $y$. This implies that for any $x$, we have $f(x) + f(x) = g(x, x) = 0$. It follows that $f(x) = 0$ for any $x$. Now consider some $x$ and $y$ so that $x \neq y$. We must have $f(x) + f(y) = 0 + 0 = 0$. However, $g(x, y) \neq 0$ because $x \neq y$. Hence, there exists $x$ and $y$ so that $f(x) + f(y) \neq g(x, y)$. We have reached a contradiction.

## G  Experiments: Data and Training Details

### G.1  Fantastic Treasure: Maximum Value Difference

**Dataset generation.**  In the dataset, we sample $50,000$ training data, $5,000$ validation data, and $5,000$ test data. For each model, we report the test accuracy with the hyperparameter setting that achieves the best validation accuracy. In each training sample, the input universe consists of 25 treasures $X_1, ..., X_{25}$. For each treasure $X_i$, we have $X_i = [h_1, h_2, h_3]$, where the location $h_1$ is sampled uniformly from $[0..20]^8$, the value $h_2$ is sample uniformly form $[0..100]$, and the color $h_3$ is sampled uniformly from $[1..6]$. The task is to answer what the difference is in value between the most and least valuable treasure. We generate the answer label $y$ for a universe $S$ as follows: we find the the maximum difference in value among all treasures and set it to $y$. Then we make the label $y$ into one-hot encoding with $100 + 1 = 101$ classes.

**Hyperparameter setting.**  We train all models with the Adam optimizer, with learning rate from $1e-3, 5e-4$, and $1e-4$, and we decay the learning rate by $0.5$ every 50 steps. We use cross-entropy loss. We train all models for 150 epochs. We tune batch size of 128 and 64.

For GNNs and HRN, we choose the hidden dimension of MLP modules from 128 and 256. For DeepSet and MLP, we choose the hidden dimension of MLP modules from 128, 256, 2500, 5000. For the MLP and DeepSet model, we choose the number of of hidden layers for MLP moduels from 4 and 8, 16. For GNN and HRN, we set the number of hidden layers of the MLP modules to 3, 4. Moreover, dropout with rate $0.5$ is applied before the last two hidden layers of $\text{MLP}_1$, *i.e.*, the last MLP module in all models.

### G.2  Fantastic Treasure: Furthest Pair

**Dataset generation.**  In the dataset, we sample $60,000$ training data, $6,000$ validation data, and $6,000$ test data. For each model, we report the test accuracy with the hyperparameter setting that

achieves the best validation accuracy. In each training sample, the input universe consists of 25 treasures $X_1, ..., X_{25}$. For each treasure $X_i$, we have $X_i = [h_1, h_2, h_3]$, where the location $h_1$ is sampled uniformly from $[0..20]^8$, the value $h_2$ is sample uniformly form $[0..100]$, and the color $h_3$ is sampled uniformly from $[1..6]$. The task is to answer what are the colors of the two treasure that are the most distant from each other. We generate the answer label $y$ for a universe $S$ as follows: we find the pair of treasures that are the most distant from each other, say $(X_i, X_j)$. Then we order the pair $(h_3(X_i), h_3(X_j))$ to obtain an ordered pair $(a, b)$ with $a \leq b$ (aka. $a = \min\{h_3(X_i), h_3(X_j)\}$ and $(b = \max\{h_3(X_i), h_3(X_j)\})$, where $h_3(X_i)$ denotes the color of $X_i$. Then we compute the label $y$ from $(a, b)$ by counting how many valid pairs of colors are smaller than $(a, b)$ (a pair $(k, l)$ is smaller than $(a, b)$ iff i). $k < a$ or ii). $k = a$ and $l < b$). The label $y$ is one-hot encoding of the minimum cost with $6 \times (6 - 1)/2 + 6 = 21$ classes.

**Hyperparameter setting.** We train all models with the Adam optimizer, with learning rate from $1e-3, 5e-4$, and $1e-4$, and we decay the learning rate by $0.5$ every 50 steps. We use cross-entropy loss. We train all models for 150 epochs. We tune batch size of 128 and 64.

For the MLP and DeepSet model, we choose the number of of hidden layers of MLP modules from 4 and 8, 16. For GNN and HRN models, we set the number of hidden layers of the MLP modules from 3 and 4. For DeepSet and MLP models, we choose the hidden dimension of MLP modules from 128, 256, 2500, 5000. For GNNs and HRN, we choose the hidden dimension of MLP modules from 128 and 256. Moreover, dropout with rate 0.5 is applied before the last two hidden layers of $\text{MLP}_1$, *i.e.*, the last MLP module in all models.

### G.3 Monster Trainer

**Task description.** We are a monster trainer who lives in a world $S$ with 10 monsters. Each monster $X = [h_1, h_2]$ has a location $h_1 \in [0..10]^2$ and a unique combat level $h_2 \in [1..10]$. In each game, the trainer starts at a random location with level zero, $X_{\text{trainer}} = [p_0, 0]$, and receives a quest to defeat the level-$k$ monster. At each time step, the trainer can challenge any *more powerful* monster $X$, with a cost equal to the product of the travel distance and the level difference $c(X_{\text{trainer}}, X) = \|h_1(X_{\text{trainer}}) - h_1(X)\|_{\ell_1} \times (h_2(X) - h_2(X_{\text{trainer}}))$. After defeating monster $X$, the trainer's level upgrades to $h_2(X)$, and the trainer moves to $h_1(X)$. We ask the minimum cost of completing the quest, *i.e.*, defeating the level-$k$ monster. The range of cost (number of classes for prediction) is 200. To make games even more challenging, we sample games whose optimal solution involves defeating three to seven non-quest monsters.

**A DP algorithm for shortest paths that needs half of the iterations of Bellman-Ford.** We provide a DP algorithm as the following. To compute a shortest-path from a source object $s$ to a target object $t$ with at most seven stops, we run the following updates for four iterations:

$$\text{distance}_s[1][u] = \text{cost}(s, u), \qquad \text{distance}_s[k][u] = \min_v \big\{\text{distance}_s[k-1][v] + \text{cost}(v, u)\big\}, \tag{G.1}$$

$$\text{distance}_t[1][u] = \text{cost}(u, t), \qquad \text{distance}_t[k][u] = \min_v \big\{\text{distance}_t[k-1][v] + \text{cost}(u, v)\big\}. \tag{G.2}$$

Update Eqn. G.1 is identical to the Bellman-Ford algorithm Eqn. 4.2, and $\text{distance}_s[k][u]$ is the shortest distance from $s$ to $u$ with at most $k$ stops. Update Eqn. G.2 is a *reverse* Bellman-Ford algorithm, and $\text{distance}_t[k][u]$ is the shortest distance from $u$ to $t$ with at most $k$ stops. After running Eqn. G.1 and Eqn. G.2 for $k$ iterations, we can compute a shortest path with at most $2k$ stops by enumerating a mid-point and aggregating the results of the two Bellman-Ford algorithms:

$$\min_u \big\{\text{distance}_s[k][u] + \text{distance}_t[k][u]\big\}. \tag{G.3}$$

Thus, this algorithm needs *half of the iterations of Bellman-Ford*.

**Dataset generation.** In the dataset, we sample $200,000$ training data, $6,000$ validation data, and $6,000$ test data. For each model, we report the test accuracy with the hyperparameter setting that achieves the best validation accuracy. In each training sample, the input universe consists of the trainer and 10 monsters $X_0, ..., X_{10}$, and the request level $k$, *i.e.*, we need to challenge monster $k$. We have $X_i = [h_1, h_2]$, where $h_1 = i$ indicates the combat level, and the location $h_2 \in [0..10]^2$ is sampled uniformly from $[0..10]^2$. We generate the answer label $y$ for a universe $S$ as follows. We implement a shortest path algorithm to compute the minimum cost from the trainer to monster $k$,

where the cost is defined in task description. Then the label $y$ is a one-hot encoding of minimum cost with 200 classes. Moreover, when we sample the data, we apply rejection sampling to ensure that the minimum cost's shortest path is of length $3, 4, 5, 6, 7$ with equal probability. That is, we eliminate the trivial questions.

**Hyperparameter setting.** We train all models with the Adam optimizer, with learning rate from $2e-4$ and $5e-4$, and we decay the learning rate by $0.5$ every $50$ steps. We use cross-entropy loss. We train all models for $300$ epochs. We tune batch size of $128$ and $64$.

For the MLP model, we choose the number of layers from $4$ and $8, 16$. For other models, we choose the number of hidden layers of MLP modules from $3$ and $4$. For GNN models, we choose the hidden dimension of MLP modules from $128$ and $256$. For DeepSet and MLP models, we choose the hidden dimension of MLP modules from $128, 256, 2500$. Moreover, dropout with rate $0.5$ is applied before the last two hidden layers of MLP$_1$, *i.e.*, the last MLP module in all models.

### G.4 SUBSET SUM

**Dataset generation.** In the dataset, we sample $40,000$ training data, $4,000$ validation data, and $4,000$ test data. For each model, we report the test accuracy with the hyperparameter setting that achieves the best validation accuracy. In each training sample, the input universe $S$ consists of 6 numbers $X_1, ..., X_6$, where each $X_i$ is uniformly sampled from [-200..200]. The goal is to decide if there exists a subset that sums up to $0$. In the data generation, we carefully decrease the number of questions that have trivial answers: 1) we control the number of samples where $0 \in \{X_1, ..., X_6\}$ to be around 1% of the total training data; 2) we further control the number of samples where $X_1 + ... + X_6 = 0$ or $\exists i, j \in [1..6]$ so that $X_i = -X_j$ to be around 1.5% of the total training data. In addition, we apply rejection sampling to make sure that the questions with answer yes (aka. such subset exists) and answer no (aka. no such subset exists) are balanced (*i.e.*, 20,000 samples for each class in the training data).

**Hyperparameter setting.** We train all models with the Adam optimizer, with learning rate from $1e-3, 5e-4$, and $1e-4$, and we decay the learning rate by $0.5$ every $50$ steps. We use cross-entropy loss. We train all models for $300$ epochs. The batch size we use for all models is $64$.

For DeepSets and MLP models, we choose the number of of hidden layers of the MLP modules from $4, 8, 16$. For GNN and HRN models, we set the number of hidden layers of the last MLP modules to $4$. For DeepSets and MLP, we choose the hidden dimension of MLP modules from $128, 256, 2500, 5000$. For GNN and HRN models, we choose the hidden dimension of MLP modules from $128$ and $256$. Moreover, dropout with rate $0.5$ is applied before the last two hidden layers of MLP$_1$, *i.e.*, the last MLP module in all models.

The model Neural Exhaustive Search (NES) enumerates all possible non-empty subsets $\tau$ of $S$, and passes the numbers of $\tau$ to an MLP, in a random order, to obtain the hidden feature. The hidden feature is then passed to a single-direction one-layer LSTM of hidden dimension $128$. Afterwards, NES applies an aggregation function to these $2^6 - 1$ hidden states obtained by the LSTM to obtain the final output. For NES, we set the number of hidden layers of the last MLP, *i.e.*, MLP$_2$, to $4$, the number of hidden layers of the MLPs prior to the last MLP, *i.e.*, MLP$_1$, to $3$, and we choose the hidden dimension of all MLP modules from $128$ and $256$.

