# OpenReview forum: "What Can Neural Networks Reason About?"
_ICLR.cc/2020/Conference — Accept (Spotlight)_

### Official Review · AnonReviewer3 · 2019-10-21
**Official Blind Review #3**

**Rating:** 8

**Review:**

The paper proposes a measure of classes of algorithmic alignment that measure how "close" neural networks are to known algorithms, e.g. dynamic programming (DP). The measure is based on the number of samples needed such that the expected generalization error is less than epsilon with 1-delta probability, where epsilon and delta are free parameters.

The paper proves the link between several classes of known algorithms and neural network architectures by showing how their sample complexity varies. For instance the paper shows that Graph Neural Network (GNN), can approximate any DP algorithm in a sample efficient manner, whereas MLP and deep sets (permutation invariant NN) can't. The paper empirically verifies their claims on 4 toy datasets, each representing an increasingly complex algorithm needed to solve the problem.

I recommend this paper be accepted, since I think it's an important direction of research, and it formalizes a lot of intuition about neural network architectures.

It would be very interesting if the authors could actually compute the number of samples, M, for different NN architectures on the toy datasets, and show how it matches empirical findings. This could be a powerful tool if it could be made easy to use for the common practitioner.

**Experience Assessment:**

I have published one or two papers in this area.

**Review Assessment: Checking Correctness Of Derivations And Theory:**

I assessed the sensibility of the derivations and theory.

**Review Assessment: Checking Correctness Of Experiments:**

I assessed the sensibility of the experiments.

**Review Assessment: Thoroughness In Paper Reading:**

I read the paper at least twice and used my best judgement in assessing the paper.

---

> ### Author Response · Authors · 2019-11-07
> **Response**
>
> Thank you for appreciating our work and giving the nice suggestion. It would indeed be very interesting to see sample sizes for different architectures and tasks in practice. However, the number of samples needed for models like MLP to learn the more complex tasks, e.g. DP, would be very high, so the experiments will be prohibitively expensive. We are considering experimenting models on smaller training set and plot  accuracy v.s. sample size to showcase the trend. We have included the experimental results in the revised version (Fig 4 and Sec 4.3).

---

> > ### Comment · AnonReviewer3 · 2019-11-14
> > **Update**
> >
> > Thank you for adding the experiments.
> >
> > I've decided to retain my original rating.

---

### Official Review · AnonReviewer1 · 2019-10-21
**Official Blind Review #1**

**Rating:** 6

**Review:**

This work seeks theoretical and empirical proof of the reasoning capacity of neural networks. The authors build on a body of research that demonstrates the usefulness of different neural network architectures for different reasoning problems. For example, Deep Sets have been proposed to answer questions about sets (e.g., a summary statistic), and GNNs about graph related problems, such as shortest path.

I anticipate that readers would be very satisfied with the intuition behind the main result: neural networks that “align” with known algorithmic solutions are better able to learn the solutions. Many architectures have been proposed over the years, often with a high-level justification for the architecture’s form. For example, Relation Networks noted the difficulty with learning n^2 relations using an MLP, which is an observation reflected in this work’s explanation of the difficulty with learning a for loop.

Provided here is a justification for these high-level design decisions. The authors provide some theory and experimental results to demonstrate their proposed notion of alignment, and show that NNs that align with known algorithmic solution do well, while those that do not align do not do well. In particular, I appreciate both the positive and negative evidence, since demonstrating lack of alignment (and poor performance) is a necessary condition to show alongside alignment (and good performance).

I’d like to caution the authors regarding their main conclusion, which is stated a few times in the paper:

“This perspective suggests that whether a neural network can learn a reasoning task depends on whether there exists an algorithmic solution that the network aligns with”.

I think this logic is not precisely correct, and I would modify this to:

“If the structure of a neural network aligns with a known algorithmic solution, then it can more easily learn a reasoning task than a neural network does not align”.

This is a subtle but important difference. In particular, the original logic does not capture situations where an algorithmic solution is not known, but a neural network can otherwise still learn a solution (consider object classification). I think even the corrected logic as I’ve spelled it out above might not be quite right either, since it does not consider situations where the algorithmic solution exists, but it obtuse. Would a neural network easily learn such a task?

Overall I think the paper is clearly written, and the experiments are adequate. Unfortunately I am not well-versed in the theoretical literature on this topic, so my assessment of the proofs is limited, and I will need to defer to the other reviewers on these matters. My surface level assessment of them is that the logic seems generally sound, but I cannot make any strong statements placing them in the context of previous work, nor can I properly evaluate the nuances. Nonetheless, as a whole, I think this is a strong contribution and a nicely put together piece of work.


**Experience Assessment:**

I have published one or two papers in this area.

**Review Assessment: Checking Correctness Of Derivations And Theory:**

I assessed the sensibility of the derivations and theory.

**Review Assessment: Checking Correctness Of Experiments:**

I assessed the sensibility of the experiments.

**Review Assessment: Thoroughness In Paper Reading:**

I read the paper at least twice and used my best judgement in assessing the paper.

---

> ### Author Response · Authors · 2019-11-07
> **Response**
>
> Thank you for your constructive feedback. Reviewer points out an imprecise statement/conclusion in our paper. We have adopted the reviewer's suggested version in the revised revision.
>
> Reviewer asks whether neural networks can learn tasks where the algorithm is not known. Our answer is the algorithm we hope to learn does not need to be known, but knowing the structure of the algorithmic solution can help with designing architectures and theoretical guarantees. For example, our experiments (Sec 4.1, 4.3) show that different architectures that align to different algorithms can both learn the task well.
>
> Reviewer asks to more carefully consider the situation where the algorithmic solution exists but is obtuse. In this paper, we focus on reasoning tasks whose underlying algorithm is exact and has clear structure, and leave the study of approximation algorithms (do not solve the task exactly) and unknown structures for future work. We discussed this at the end of Sec 3 at page 4. This should clarify the the range of problems we address in this paper, and how our results relate to various situations.
>
> In the case where we face a problem where we do not have knowledge about the underlying algorithmic structure, in order to still generalize well, we think neural architecture search over the algorithmic structure space could be a promising future direction. We will discuss these in the final version.

---

### Official Review · AnonReviewer2 · 2019-10-23
**Official Blind Review #2**

**Rating:** 8

**Review:**

This paper presents a framework, dubbed algorithmic alignment, based on PAC learning and sample complexity, with the aim to explain generalization on reasoning tasks for different neural architectures. The framework roughly states that in order for the model to be able to learn and successfully generalize on a reasoning task, it needs to be able to easily learn (to approximate) steps of the reasoning tasks. The authors use this framework to propose an increasingly difficult set of tasks, designed to showcase the type of models that would be fit or unfit to solve them. The resulting experiments corroborate the theory, showing the limits of MLPs, Deep Sets, and consequently Graph Neural Networks. The final claim that an NP-hard task needs an enumerative architecture, and then experimental validation of that claim is nice and fits into the theory.
The added benefit of the paper is that the authors show as a side-effect that visual question answering and intuitive physics

Overall, the paper presents a meaningful contribution to the theory of learning, formalizing the means of quantifying the capabilities of architectures to solve tasks of certain complexity. The paper, though dense, is well well written, and carries an interesting conclusion that better algorithmic alignment brings the sample complexity down, i.e. models with better algorithmic alignment to the task (function they want to approximate) should generalize better.
The formalization presented in the paper, though remarkably intuitive, might be difficult to practically use for more elaborate models and it is not clear whether it can be numerically computed. The paper (i.e. the reader) would certainly benefit from more examples of algorithmic alignment comparison of different models, such as one done in Corollary 3.7.

Question:
- difference to Kolmogorov complexity is that any algorithmic alignment that yields decent sample complexity is good enough - how do you define decent?
- You state: “in Section 4, we will show that we can usually derive a near-optimal alignment by avoiding as many “for loops” in algorithm steps as possible.” yet I did not see that there. Was that effectively shown in Corollary 3.7?

Slightly related work: On the Turing Completeness of Modern Neural Network Architectures

**Experience Assessment:**

I do not know much about this area.

**Review Assessment: Checking Correctness Of Derivations And Theory:**

I assessed the sensibility of the derivations and theory.

**Review Assessment: Checking Correctness Of Experiments:**

I assessed the sensibility of the experiments.

**Review Assessment: Thoroughness In Paper Reading:**

I read the paper at least twice and used my best judgement in assessing the paper.

---

> ### Author Response · Authors · 2019-11-07
> **Response**
>
> Thank you for your helpful feedback. We answer your questions below.
>
> - “difference to Kolmogorov complexity is that any algorithmic alignment that yields decent sample complexity is good enough - how do you define decent?”. Here, “decent” is a loose term we use to refer to a tight enough algorithmic alignment for good generalization performance. We will explain more in the revised version.
>
> - “You state: ‘in Section 4, we will show that we can usually derive a near-optimal alignment by avoiding as many ‘for loops’ in algorithm steps as possible.’ yet I did not see that there”. One example is Section 4.2: DeepSets does not algorithmically align well with the relational argmax task. It has to learn the for-loops (Claim 4.1), which requires many samples. On the other hand, GNN algorithmically aligns well with relational argmax --- the for-loops are hard-coded in the computation graph. Therefore, GNN achieves better sample efficiency by avoiding learning the for-loop. We will make the connection clearer in the revised version.
>
> - We will add the suggested reference and discuss the relation in the revised version.

---

> > ### Comment · AnonReviewer2 · 2019-11-14
> > **My update**
> >
> > I’ve checked the changes in the paper and I’ve read the correspondence between reviewers and the authors in detail. In particular, I am very grateful to Hao Tang for his involvement in the process and his questions and authors’ replies clarified a few things for me. The addition of sampled training data experiments is a welcome addition to the paper and a good spot by reviewer #3. The whole discussion made my understanding of the paper clearer and I’m happy to increase my score as I think the community will profit from further development of theory explaining generalizations of different NN architectures, especially when well experimentally supported. The presented paper does solid work on this for reasoning tasks, and in my opinion it warrants a publication.

---

### Public Comment · ~Hao_Tang5 · 2019-11-06
**Concerns about the theoretical part (sample complexity bound, def. of algorithm alignment etc.)**

Dear authors,

Thanks for sharing the work. In my understanding, the paper is aiming at formalizing the relational inductive bias intuition in [1] into a more concrete concept (the algorithm alignment), theoretically proving the advantages of GNNs, and experientially evaluating the claim.
I believe in the intuition about relational inductive bias and that GNNs are truly more sample efficient than MLP on many relation-related and reasoning-related tasks. And I won't disagree with that the algorithm alignment is a promising direction of formalizing GNNs' relational inductive bias. The experimental part of this paper does show some promising results aligned with those intuitions and provides some analysis of GNNs' power on learning different algorithms according to the experimental results. The paper is overall an interesting paper even without the theoretical part.

However, I do have some concerns about the theoretical part of this paper.   Most importantly, based on a very strong assumption (Sequential learning in Theorem 3.6.), the sample complexity bound (Theorem 3.6.) and then the algorithm alignment definition (Definition 3.4.) proposed in the paper are somehow restrictive and counter-intuitive. The strong assumption not only makes the comparison between MLP and GNNs' sample complexity unfair but also may mislead GNNs' architecture design in the future.
1. The sequential learning assumption is a very strong assumption even in the field of PAC learning etc.. It assumes oracles that can supervise each MLPs' behaviours in the neural networks. Actually, MLPs are all composed of several MLPs. If such oracles are available for MLPs, the sample complexity bound of MLPs would be the same as or even lower than that of GNNs. The comparison between GNN and MLP's sample complexity is therefore unfair. Also, it is kind-of inaccurate to compare the sample complexity bound to support some claims in the paper, although the intuitions are reasonable.
2.  The algorithm alignment definition, which is induced by the bound analysis in Theorem 3.6, is somehow counter-intuitive. $n\cdot \max_iC_{A_i}(f_i, \epsilon, \delta)\le M$. For example, the sample complexity scales linearly with the MLP modules' number $n$. It induces that increasing the depth of GNNs will not have a huge or dramatic influence on the generalizability or sample efficiency, which is counter-intuitive.

I would suggest the authors to put the strong assumption in a more conspicuous position to avoid misunderstandings of readers (e.g. in the introduction). Otherwise, the authors could relax the bound or simply justify the intuition according to some results in the cognitive science field. It is still interesting to see that GNNs are experientially good at learning DP and the other experimental results as well.

Some less important concerns are listed:
1. It may be better to show some failure cases of GNNs. For example, it's reasonable to see GNN7 behaves poorly on (a) and (b) tasks in Figure 3.
2. Sample complexity bound $\neq$ sample complexity. Maybe should be less confident while comparing MLPs and GNNs.
3. The connections between problem/task-alignment and algorithm-alignment are not clear enough.
4. The first paragraph in Section 3 could be more accurate. For example, the performance difference of different modules may come from the data quality, optimizer and hyper-parameter tuning ability etc...

[1] Battaglia, Peter W., et al. "Relational inductive biases, deep learning, and graph networks." arXiv preprint arXiv:1806.01261 (2018).

---

> ### Author Response · Authors · 2019-11-07
> **Response**
>
> Thank you for your interest in our work. We address your concerns below.
>
> While we agree that assumptions of our theorems are strong, we do not over-claim: We have clearly stated our assumptions in the paper and discussed the relation to practice (Sec 3.2). We also write in the introduction that we provide “initial theoretical support” to show that algorithmic alignment is desirable for generalization under “simplifying assumptions”. Several theoretical works on deep learning at times make simplifying assumptions. Still, these works have led to interesting insights and triggered many follow-up works. The main goal of our paper is to introduce the perspective of algorithmic alignment and take the first formal initiative towards understanding the interplay of reasoning tasks and NN architecture. Moreover, as we have discussed in Sec 3.2, in our experiments, all models are trained end-to-end. The experimental results agree with our theoretical results despite our sequential assumption, so we believe that future work can extend our theoretical results to end-to-end learning.
>
>
> However, we strongly disagree with the reader’s other concerns.
>
> - “If such oracles are available for MLPs, the sample complexity bound of MLPs would be the same as or even lower than that of GNNs. The comparison between GNN and MLP's sample complexity is therefore unfair.” This is not correct. Our comparison is fair. Although we assume an oracle for each sub-module in Thm 3.6, we do *not* assume oracles for individual layers in the MLP modules of GNN. If we add oracles to every layer of both the MLP and each MLP module in GNN as the reader suggests, we can still show that GNN has a better sample complexity. Intuitively, this is because the giant MLP still needs to learn the entire for-loop. On the other hand, GNNs do not need to learn the for-loop because it is encoded in the architecture (Fig. 2). In our theorem, we try our best to keep the number of oracles small so that it is close to practice, where models are trained end-to-end. Therefore, we do not assume oracles in MLP layers. Also, our theorem agrees with experimental results, so future work may further relax the assumption.
>
> - “[Our theorem] induces that increasing the depth of GNNs will not have a huge or dramatic influence on the generalizability or sample efficiency, which is counter-intuitive. ” This is not correct. Increasing the depth of GNNs is crucial to achieving better algorithmic alignment for some tasks (Sec 4.3) and therefore improving sample efficiency. If the depth of the GNN is not sufficient, at least one of the sub-modules needs to learn for-loops. But GNNs with more iterations can align better and avoid such for-loops. One example is the shortest paths problem [Figure 3c]: The number of GNN iterations is the key to good performance. For other tasks, e.g. Fig 3ab, increasing GNN depth is not so necessary. Based on our theory and experiments, both GNN1 and GNN3 can perform well on simple relational argmax tasks. We hope this clarifies your concerns.
>
> - “I believe in the intuition about relational inductive bias and that GNNs are truly more sample efficient than MLP on many relation-related and reasoning-related tasks.” We would like to clarify that our intuition is more specific than what the reader describes. We not only formalize the relational inductive bias of some popular reasoning architectures, but we also characterize *which tasks* GNN does well, and provide examples where GNN fails.
>
> Reply to minor concerns:
>
> - “It may be better to show some failure cases of GNNs.” We have shown a failure case in the paper --  GNNs fail on the subset-sum task in Fig 3(d), while NES, an architecture that aligns better with the task, generalizes well.
>
> - “It's reasonable to see GNN7 behaves poorly on (a) and (b) tasks, i.e. summary statistics and relational argmax, in Figure 3.” This is not correct. GNN7 performed well in our experiments (we do not show performances of all GNN depths in paper due to space limit).

---

> > ### Public Comment · ~Hao_Tang5 · 2019-11-08
> > **Still confused**
> >
> > Dear authors,
> >
> > Thanks for the detailed response. And please feel free to correct me. It is one of the advantages of OpenReview that our readers can directly consult the authors.
> >
> > I have read your reply carefully. But I still have a few concerns about the theoretical part.
> > 1. In my understanding, a fair comparison should give both MLP and GNN the same power of oracles. And then, I tried to derive the sample complexity bound of MLP given your oracle. In my understanding, the sample complexity of MLPs is not high compared to GNNs when approximating the sum pooling, e.g. “How many objects are either small cylinders or red things?” in Sec. 4.1. (Your for-loop argument for sum pooling is still confusing for me). However, it would be difficult for MLPs to approximate max pooling especially when generalizability is considered. So, does the "for-loop" increase the sample complexity? Or it's the max pooling that the MLPs are difficult to approximate in a sample efficient way?
> > From another perspective, in my understandings, many GNN variants are shared-weight and highly-regularized MLPs, such as GCN, GIN, and your GNN in Sec.2 (without softmax). For a fixed-dimensional input, given a GCN/GIN/your-GNN, it's easy to construct a MLP that will perform exactly the same on any input. And the key to those GNN variants' sample efficiency is those constraints from GNNs' architecture that restrict or apply a strong prior distribution of the solution space. However, the oracles can apply similar constraints to a general MLP by approximating their intermedia output. Therefore, the sample complexity of GNNs+oracle and MLP+oracle should be similar. This is my original point. I wasn't aware of the difficulty of approximating max pooling by MLPs, although.
> > (An assumption utilized in these arguments is that the sample complexity to approximate sum/mean pooling by MLP is not high compared to that to approximate a single step in DP. It aligns with my intuition, but please feel free to correct me. I will also do some experiments later. The topic is interesting for me anyway.)
> > 2. Since the paper is talking about the generalizability and sample efficiency, I assume overfitting is a related topic. According to your sample complexity bound formula, 2k-layer GNNs just need twice as many data samples as k-layer GNNs to achieve similar generalizability for the same DP problem for any sufficiently large k (a simple proof can be found later). This is where I found the formula counter-intuitive. In my understanding, the generalizability or sample complexity of GNNs should generally scale at least quadratically or even exponentially with the layer number without assumptions about residual connections or normalization layers. If your sample complexity bound is reasonable, it would be a very strong and useful conclusion at least for me. Regardless of the theoretical part, if the authors could show the conclusion experientially by comparing the test-error-distribution with respect to the data sample numbers for GNNs of different layer numbers, it would be still a very interesting contribution.
> > The failure cases of GNNs are also referred to those Deep GNNs. It's reasonable to see GNN9-30 overfitting and therefore less generalizable.
> >
> > As stated in my first comment, I am not underrating the paper. Instead, I think the paper does a good job in making the abstract concept, relational inductive bias, concrete from different perspectives as stated by the authors. The experimental results also align with the analysis. The insight could be helpful in the future. But I hope that the theoretical part could be more accurate to avoid misunderstandings.
> >
> > ================= some proofs ==================
> > Claim: According to your sample complexity bound formula, 2k-layer GNNs just need twice as many data samples as k-layer GNNs to achieve similar generalizability for the same DP problem for any sufficiently large k.
> > Proof: For any DP problem, there will be a layer number, $k$, that is large enough so that each MLP module can just learn a simple function and therefore the maximum sample complexity of MLP submodules is bounded (I think this is also assumed in the paper). Then, for the $2k$-layer GNNs, the $k+1$ to $2k$th GNN layers just need to learn the identity functions, which could be easy for many GNN variants (e.g. the GAT variants). Therefore, the $\max_i C_{A_i}(f_i, \epsilon, \delta)$ part won't change much and the sample complexity bound scales linearly.

---

> > > ### Author Response · Authors · 2019-11-10
> > > **Confusion comes from reader’s incorrect assumption**
> > >
> > > Thanks for your interest again. We clarify your confusion below.
> > >
> > > 1. The reader is confused because the reader’s assumption -- “the sample complexity to approximate sum/mean pooling by MLP is not high compared to that to approximate a single step in DP.” is indeed not correct. For-loops, including sum/max over functions of all objects, have high sample complexity for an MLP to learn by Thm 3.5, compared to a single step in DP, which is usually a function on a pair of objects (e.g., Bellman-Ford relaxation in Fig. 2). We also discuss this in Sec 3.1, 3.2, and show an example with sum-pooling in Corollary 3.7, where sample complexity increases polynomially with the number of objects to loop over. GNNs could avoid learning such for-loops in DP algorithms so they generalize well. Although Thm 3.5 also has simplifying assumptions, e.g. using gradient descent with infinitesimally small steps, it aligns well with our experimental results (Fig.3).
> > >
> > > 2. This is a good question. Indeed, our bound suggests reasonably deep GNN should generalize well, even if its number of iterations is higher than the DP iteration. As the reader suggests, we have run additional experiments with GNN10 (each sub-module is a 4-layer MLP) on summary statistics task, where GNN1 already performs well. Our experiment shows GNN10 performs equally well as GNN1. Thus, the experiment aligns with our theory here. We also found this result interesting and will expand on it a bit more in the final version. For example, it contrasts with what has been observed in GNN node classification tasks on social networks etc [1], where without JK, 2-layer GNN often perform the best and deeper GNN perform worse. There are several differences between our settings and theirs, one being adaptivity (different algorithm steps and number of steps we shall act on each node) is often needed for different nodes depending on subgraph structures (expanders vs. trees) in node classification tasks [1], which is not the case in many reasoning tasks. Moreover, note that our GNN formula for reasoning (Eqn 2.2) is different from GCN and GIN, e.g. GCN uses one-layer perceptron but our reasoning GNN (Eqn 2.2) uses MLP. Our reasoning GNN also explicitly models pairwise functions but GCN and GIN do not. This makes a difference too, so for failures of deep GCN on node classification tasks do not necessarily hold here.
> > >
> > > [1] Representation learning on graphs with jumping knowledge networks. ICML 2018.
> > >
> > > Hopefully this clarifies your confusion. Please let us know if you have other questions. Again, we appreciate your interests.

---

### Author Response · Authors · 2019-11-07
**General response**

We sincerely appreciate all the reviews, they give positive and high-quality comments on our paper with a lot of constructive feedback. We answer each reviewer’s questions individually. We will update the draft soon.

---

### Author Response · Authors · 2019-11-13
**Update**


Dear Reviewers and AC,

We have updated our draft to incorporate the nice suggestions of the reviewers. In particular, we have made the following changes:

- We have added additional experiments to show test accuracy v.s. training set size on sub-sampled training sets to further support our theory. The results are shown on Figure 4 at page 7, which is also discussed in Sec 4.3. We thank Reviewer 3 for the good suggestion of probing the effect of the number of samples empirically.
- Thanks to Reviewer 1 for a helpful comment that points out an imprecise statement. We changed it to make it more accurate, and added a discussion at the end of Sec 3 (page 4) regarding reasoning algorithms whose structure is obtuse, and regarding approximation algorithms. This should clarify the the range of problems we address in this paper, and how our results relate to various situations.
- We have added the related work as suggested by Reviewer#2.
- We will improve other minor points in the final version.

In addition, we have clarified all the concerns and confusion of the public comment regarding our theoretical parts.

Please let us know if you have additional questions.

Thank you,
Authors

---

> ### Comment · AnonReviewer1 · 2019-11-14
> **Thanks**
>
> Thank you for your updates. I am satisfied with the quality of this work and I recommend its acceptance.

---

### Decision · Program_Chairs · 2019-12-19

**Decision:**

Accept (Spotlight)

**Comment:**

This paper proposes a framework which qualifies how well given neural architectures can perform on reasoning tasks. From this, they show a number of interesting empirical results, including the ability of graph neural network architectures for learn dynamic programming.

This substantial theoretical and empirical study impressed the reviewers, who strongly lean towards acceptance. My view is that this is exactly the sort of work we should be show-casing at the conference, both in terms of focus, and of quality. I am happy to recommend this for acceptance.